# Epidemiological and Mycological Aspects of Onychomycosis in Dakar (Senegal)

**DOI:** 10.3390/jof5020035

**Published:** 2019-04-29

**Authors:** Khadime Sylla, Roger C. K. Tine, Doudou Sow, Souleye Lelo, Mamadou Dia, Seyda Traoré, Babacar Faye, Thérèse Dieng

**Affiliations:** 1Department of Parasitology-Mycology, Faculty of Medicine, Pharmacy and Odontology, University Cheikh Anta Diop, Dakar 5005, Senegal; rogertine@hotmail.com (R.C.K.T.); doudsow@yahoo.fr (D.S.); lelosouleye82@hotmail.com (S.L.); seydatraore04@yahoo.fr (S.T.); bfaye67@yahoo.fr (B.F.); therese.dieng@ucad.edu.sn (T.D.); 2Laboratory of Parasitology-Mycology, National University Hospital of Fann, Dakar 5035, Senegal; oudamamdia@gmail.com

**Keywords:** onychomycosis, epidemiology, yeasts, dermatophytes, molds, Senegal

## Abstract

Onychomycosis is a fungal nails infection often caused by yeasts, dermatophytes and molds. It is an important public health concern due to its high prevalence, the problem of diagnostics, and the poor response to treatments. The objective of this study was to evaluate the epidemiological and microbiological profile of onychomycosis diagnosed at the Laboratory of Parasitology-Mycology of the National University Hospital of Fann in Dakar, Senegal, from 2012 to 2016. A retrospective and descriptive study was performed from January 2012 to December 2016 in a patient attending the laboratory of Parasitology-Mycology at the Fann teaching hospital. Socio-demographic, clinical and biological data were collected from the bench registers. Samples from the lesions were tested using direct microscopy and cultured on a Sabouraud-Chloramphenicol and Sabouraud-Chloramphenicol-Actidione medium. A descriptive analysis was done using Stata IC 12 software. The significance level of different tests was set at 5% two-side. A total of 469 patients were included in this study. The mean age of the study population was 33.2 ± 15.2 years, and the sex ratio was 0.52. The prevalence of onychomycosis was 48.4% (227/469). The main clinical presentations were disto-lateral subungual onychomycosis (37.9%) and onyxis (46.5%). Identified fungal species were *Candida albicans* (42.7%), *Candida* spp. (39.5%), *Trichophyton soudanense* (10.1%), *Fusarium* spp. (5.3%), and *Candida tropicalis* (2.6%). *Candida albicans* was more frequent in subjects over 15 years of age (43.6%) and women (45%). However, *Trichophyton soudanense* was higher in patients under 15 years old (17.4%) as well as in male subjects (18.8%). In conclusion, onychomycosis is a common cause of consultation in health facilities. *Candida albicans* and *Trichophyton Soudanense* are the main fungal species causing onychomycosis. A better understanding of the epidemiology of onychomycosis as well as the spectrum of the pathogen could contribute to improve the management of the infection.

## 1. Introduction

Onychomycosis is a common denomination used to describe a nail infection usually caused by three groups of fungi, namely dermatophytes, non-dermatophytes mold and yeast. Onychomycosis affects approximatively between 8 and 10% of the general population [1] and represents around 30% of all superficial fungal infections [2] and 50% of nails disorder [3]. It is even more important in the elderly [1].

The consequences of onychomycosis are mainly aesthetic from the patient’s point of view. By causing a deformation of the nail tablet, the consequences can be associated with some complications. In diabetics, they could be a risk factor for plantar perforators, and could indirectly cause the recurrence of erysipelas by recontaminating the interdigitated plantar intertrigo. In addition, local pain and irritation may be associated, as well as limited walking faculties or other limited physical practices [1,4].

The clinical symptoms are various, but it should be noted that a clinical examination is generally insufficient to differentiate a dermatophyte or *Candida* onychomycosis from another onychopathy (psoriatic, traumatic, etc.) or onychomycosis due to mold. Several factors, such as the microorganism, clinical presentation, severity of the infection, and comorbidities, should be taken into consideration when determining the best therapeutic strategy for onychomycosis [5,6].

About half of onychopathies are of fungal origin, and a mycological examination is essential to confirm the diagnosis of onychomycosis, identify the fungal species, and to follow the efficacy of an antifungal treatment [7].

The treatment of onychomycosis can be difficult, and it depends on the etiological agent. In addition, the cost of the treatment can be expensive, and some protocols may be too long, hence the need to diagnosis onychomycosis [8].

In Senegal, onychomycosis is a frequent cause of morbidity in health facilities, but its frequency and impact on the health of populations is not well assessed. Recent studies conducted in Dakar showed that onychomycosis prevalence is very important, from 44 to 58% [9,10].

A better understanding of the epidemiology of onychomycosis and the spectrum of pathogens, could contribute to improve the case management of the disease. Thus, we carried out this study aimed at evaluating the epidemiological and microbiological profile of onychomycosis diagnosed at the Laboratory of Parasitology-Mycology of the National University Hospital of Fann in Dakar, Senegal from 2012 to 2016.

## 2. Materials and Methods

### 2.1. Study Design and Population

A retrospective and descriptive study was carried out from January 2012 to December 2016 in all patients attending the Laboratory of Parasitology-Mycology of Fann University Hospital Center for a mycological examination to confirm onychomycosis.

### 2.2. Data Collection

Sociodemographic, clinical and biological data from patients were collected using the laboratory records. The following variables were collected: year and month where the samples were collected; age and sex of the patients; the clinical presentation of the lesion and biological results. The age was defined in 2 categories (less than 15 years and more than 15 years). The season was defined in both the dry season (October to June) and rainy season (July to September). 

### 2.3. Mycological Examination

The samples were collected before any antifungal treatment and after cleaning the affected area of the nail with alcohol at 70%. The samples were obtained from the deepest part, the underside of the nail, using a sterile nail clipper and scalpel. The nail fragments obtained were collected in sterile Petri dishes sealed with scotch tape and then sent to the laboratory for mycological analyzes. In case of perionyxis, the pus was also collected using a sterile cotton swab.

A proportion of the collected nail samples was digested by 30% potassium hydroxide (KOH). Then, the nail sample was deposited on a glass slide object with a drop of lightening liquid before being examined carefully using light microscopy (objective 10 and 40).

The nail specimens were also cultured, both on a Sabouraud-Chloramphenicol and on a Sabouraud-Chloramphenicol-Actidione medium. The tubes were incubated at room temperature (25–30 °C). The culture was observed every 24 h to detect a possible growth of the crop. The identification was based on macroscopic and microscopic examinations of cultures performed for dermatophytes and molds. In addition to the macroscopic and microscopic aspects, a confirmation of *Candida* species was based on the observation of pseudomycelium under a light microscopy, as well as being based on a germ tube test, the chlamydo-spore formation and the AUXACOLOR rapid identification system (Biorad, France).

### 2.4. Data Entry and Analysis

The data was entered using Excel software and the analysis was done with STATA IC 12 software. The quantitative variables were described in terms of the means and standard deviation. The inter group comparisons were performed using an ANOVA test or Student t test after checking the conditions of application of these tests. When these tests were not applicable, the non-parametric tests (Man Withney, Kruskall Wallis) were used. For the descriptive data, the percentage with the confidence interval (CI) was used to assess the prevalence of each outcome. The significance level of the different tests was 0.05 two-sided.

### 2.5. Ethical Considerations

This study was conducted in accordance with the Declaration of Helsinki. To respect the confidentiality, an identification code was assigned to each patient.

## 3. Results

### 3.1. Socio-Demographic Characteristics of Study Population

Overall, 469 patients were included in the study from 2012 to 2016, the number of patients received for a mycological examination increased from 59 to 114, with a high frequency in 2015 (123 patients). A significant variation of the number of patients attending the laboratory was observed through the months. The mean age of the study subjects was 33.17 ± 15.2 years. The study population was mainly constituted by patients aged over 15 years 89.5% (CI: 81.19–98.5). The subjects that were less than 15 years old represented 10.45%. Female participants were 65.9%, while male participants were 34.1%. Concerning the season, the proportion of patients with onychopathy was more important during the dry season, at 74.84%, compared to the rainy season, at 25.16% (Table 1).

### 3.2. Clinical Characteristics of Lesions

From the 469 patients attending the laboratory for a mycological confirmation of onychomycosis, the most common clinical presentations were onyxis (46.5%) and disto-lateral subungual onychomycosis (DLSO) (37.9%). An association onyxis (ingrowing of nail) and perionyxis was observed in 47 subjects (10%). Perionyxis and white superficial onychomycosis (WSO) represented 2.4% and 3.2%, respectively. (Table 2).

Among the study participant, 17.05% (80/469) had associated lesions. The main associations were: disto-lateral subungual onychomycosis (DLSO) and scalp ringworm (12.5%), disto-lateral subungual onychomycosis (DLSO) and palmo-plantar keratosis (18.75%), onyxis and scalp ringworm (23.8%), onyxis and intertrigo (22.5%), and onyxis and palmar-plantar keratosis (13.75%).

### 3.3. Mycological Data

According to the laboratory findings from 469 patients attending the laboratory during the study period, 227 of them were found positive, giving an 48.40% prevalence of onychomycosis. The principal pathogens groups that caused onychomycosis were represented by yeast 75.9%, dermatophytes 14.5% and don-dermatophytes 8.8% (Figure 1).

The main fungal species that were identified were *Candida albicans* (42.7%), *Candida* sp. (39.5%), *Trichophyton soudanense* (10.1%), *Fusarium* sp. (5.3%), *Candida tropicalis* (2.6%), *Trichophyton rubrum* and *Trichophyton violaceum*, each with 1.3%. The other dermatophytes species isolated from the mycological samples were *T. mentagrophytes*, *T*. *verrucosum*, *T. asahii* and *Microsporum langeronii*. The non-dermatophytes species were represented by *Onychocola* sp., *Scedosporium* sp., *Scytalidium* sp., *Fusarium solani*, *Aspergillus niger*, *Aspergillus candidus* and *Acremonium* sp. *Candida glabrata* and *Candida parapsilosis* were also identified (Table 3).

### 3.4. Variation of the Frequency of Onychomycosis According to the Socio-Demographic Characteristics of the Study Population

The prevalence of onychomycosis was higher in patients aged over 15 years old (48.6%) than in those less than 15 years old (46.9%). The difference was not significative (*p* = 0.83). Regarding the gender, onychomycosis was more frequent in female participants (51.13%) compared to male participants (43.12%). The prevalence of onychomycosis was higher during the dry season (49%) compared to the rainy season, where the prevalence was 46.6%, with no significant difference (*p* = 0.65) (Table 4).

A variation of the prevalence of onychomycosis was noted during the study period. The prevalence was higher in 2013 and 2016, respectively 61.5% and 57.9%. The lowest frequency was noted in 2012 (30.2%). However, according to the month where the sample was collected, no significance variation was observed (*p* = 0.08).

### 3.5. Variation of the Frequency of Onychomycosis Based on the Clinical Characteristics of the Lesions

The prevalence of onychomycosis in patients with onyxis was 47.7% (104/218). Among patients with disto-lateral subungual onychomycosis (DLSO), 42.7% (76/178) of them had a positive mycological result. Among subjects with white superficial onychomycosis (WSO), perionyxis and the association of onyxis and perionyxis, the positive rate of the mycological exam was 73.3%, 57.5% and 81.8% respectively (Table 5).

In subjects with disto-lateral subungual onychomycosis (DLSO) and scalp ringworm, 30% had a positive mycological test. For those with the association of DLSO and palmo-plantar keratosis, the prevalence was 53.3%. A positivity rate of 63.6% was found in subjects with onyxis and palmo-plantar keratosis. In those with the onyxis and scalp ringworm and in those with the onyxis and intertrigo association, the positivity rate was 57.9% and 50%, respectively.

### 3.6. Study of Onychomycosis Caused by Candida Albicans and Trichophyton Soudanense

Among isolated fungal species, *Candida albicans* and *Trichophyton soudanense* were the most common species 42.7% (*n* = 97) and 10.1% (*n* = 23).

According to the laboratory findings, *Candida albicans* was more frequent in subjects aged over 15 years (43.6%) compared to those less than 15 years (34.8%). However, *Trichophyton soudanense* was more frequent in subjects under 15 years old (17.4%) versus 9.7% in the other group age. Regarding the gender, the prevalence of *Candida albicans* was higher in female subjects (45%), compared to male subjects (37.7%). *Trichophyton soudanense* was more prevalent in male participants (18.8%) compared to female subjects (9.7%). Concerning the season, there was no significant variation between the frequency of *Candida albicans* and *Trichophyton soudanense* (Table 6).

Regarding the clinical aspects, the prevalence of *Candida albicans* in patients with the association onyxis/perionyxis and DLSO was 66.7% and 43.4% respectively. In those with perionyxis and onyxis, *Candida albicans* prevalence was 55.6% and 38.5%. Meanwhile *T. soudanense* was higher in subjects with onyxis (14.4%) and perionyxis (11.1%) (Table 7).

## 4. Discussion

Onychomycosis is a public health problem worldwide. It is no longer seen as an aesthetic problem but as a health concern. Knowing that 50% of onychopathies are not onychomycosis, a mycological diagnosis is important to confirm onychomycosis and to avoid non-adapted treatments. The objective of this study was to evaluate the epidemiological and mycological profile of onychomycosis diagnosed at the laboratory of Parasitology-Mycology at Fann University Hospital from 2012 to 2016.

Among 469 patients with clinical lesion in the nails, 227 (48.40%) had onychomycosis confirmed by mycological exams. Similar results have been described by other authors in Senegal. Diouf et al. and Seck et al., when assessing the epidemiological aspects of onychomycosis found an onychomycosis prevalence of 44.5% and 58.78% [9,10].

Other studies in Africa have demonstrated the endemic profile of onychomycosis. The result of our study is in line with what was found in Côte d’Ivoire by Konate et al., who showed a prevalence of 66% [11]. In Yaoundé, Kouotou et al. found a prevalence of 82.1% for onychomycosis [12]. In a study conducted in Casablanca between 2006 and 2010, Halim et al. found a prevalence of 65% [13]. Studies in Europe have also noted the endemic character of onychomycosis, but with a lower prevalence [14,15].

Our results showed that the frequency of onychomycosis increase with age. Subjects aged over 15 years were more affected (48.6%) compared to subjects under 15 years old (46.9%). This has been demonstrated by Diouf et al., who found in their study that onychomycosis was more important in adults (74.6%) compared to children (21.3%) [9]. Seck et al. showed that the frequency of onychomycosis was higher in patients aged between 15 and 65 years (90% of the study population) [10]. 

The result of this study is in line with what was found by Konate et al. in Abidjan, who demonstrated that adult persons were more affected [11]. Afene et al., when studying the clinical and mycological aspects of onychomycosis in Gabon, showed that patients aged between 20 and 60 years were more infected compared to other age groups [16]. Halim et al. showed, in Casablanca, a higher incidence of onychomycosis in adults (16-60 years) at 81% versus 4% in children (*p* < 0.001) [13]. In Iran, Soltani et al., when studying onychomycosis at the Dermatology Center in Tehran, showed that the prevalence of onychomycosis was higher in adult subjects (40.7%) and lower in subjects under 20 years (6.4%) [17].

In Europe, a study conducted by Burzykowski et al. confirms that the prevalence of onychomycosis increases with the age, in children (3%), in adults (21%) and in the elderly (45%) [18]. 

The low frequency of onychomycosis in children can be attributed to several factors, such as the difference in the structure of the nail plate, the lower exposure to trauma compared to adults, and the rapidity of the nail regrowth [19].

The results of our study showed that onychomycosis was more observed in female participants, with 51.13% compared to male subjects (43.12%). This was confirmed by studies carried out in Dakar, which showed a high prevalence of onychomycosis in women (49.4% and 71.8%) compared to 32.6% and 28.2% in men [9,10]. Other studies in Côte d’Ivoire and Gabon have described similar results with a high prevalence of onychomycosis in female subjects [11,16]. In Morocco, the results found by Halim et al. also show that the prevalence of onychomycosis is higher in female subjects (59.5%) compared to male participants (40.6%) [13].

Other studies have shown that onychomycosis is more frequent in men than in women. This has been demonstrated by Sbay et al. in Morocco (62% in male group versus 38% in group [20]) Other authors have shown similar results [21,22].

In our study, the main fungal species that were isolated were yeasts (*Candida albicans* (42.7%) and *Candida* sp. (39.5%)), dermatophytes (15%) and molds (8%). The same trends are observed in the study of Diouf et al.; yeasts accounted for 70.11%, dermatophytes 35.83% and molds 0.65% [9]. The study conducted by Seck et al. revealed the same trends, with 68.4% for *Candida albicans* and 23.5% for dermatophytes [10].

In Morocco, the main causes of onychomycosis are dermatophytes (61.46%), yeasts (25%), and molds (1.53%) [23].

In our study, *Candida albicans* was the common isolated species (42.7%), followed by *Candida* sp. (39.5%). Other yeast species were isolated (*Candida tropicalis:* 2.64%, *Candida parapsilosis*, *Candida glabrata* and *Trichosporon asahii*). A study conducted in the same structure between 2004 and 2011 found *Candida albicans* as the main identified fungal species (55.84%), followed by *Candida tropicalis* (7.79%) [9]. In Gabon, Afene et al., when studying onychomycosis, found as the main species *Candida albicans* (48.9%) and *Trichophyton soudanense* (43.3%) [16]. Konate et al. in Abidjan showed that *Candida tropicalis* (34.6%) and *Candida albicans* (30.3%) are the main species responsible for onychomycosis [11].

*C. albicans* was found as the main yeast responsible of onychomycosis (49%), mainly on the nails of the hands, in a study conducted in Morocco between 2006 and 2010 [13].

Concerning the dermatophytes, *Trichophyton soudanense* was the most isolated dermatophyte (10.1%) in our study. The other dermatophytes species were *T. rubrum* and *T. violaceum.*

The geographical distribution of dermatophytes explains the different profile of species based on the study area. In Africa, *T. soudanense* is the most predominant species, whereas in Europe *T. rubrum* is the first isolated species of both toes and hands.

In this study, only one case of *Microsporum langeronii* onychomycosis was observed. The involvement of the genus *Microsporum* is mainly observed in the scalp ringworm [7].

Pseudo-dermatophytes such as *Onychocola* sp. (0.4%) and *Scytalidium* sp. (0.4%) were also isolated in this study. Similar results were described by Hajoui et al. in Morocco; they found one case of *Onychocola canadensis* (0.7%) and one case of *Scytalidium dimitiatum* (0.7%) [24].

After the pseudo dermatophytes, other isolated mold species were represented by: *Fusarium* sp. (12 cases), one case of *Fusarium solani*, *Aspergillus niger*, *Aspergillus candidus*, *Acremonium* sp. and *Scedosporium* sp. Molds are therefore the least represented species in this series. This can be explained by the difficulty of asserting the pathogenic character of a mold which rests on a bundle of arguments including the renewal of the sampling and their isolation in a pure culture.

The limitations of the study include two major limitations:

On the bench register used to collect the data, the sampling site (hand or foot) was not specified. This would have allowed us to have an idea about the distribution of the species according to the site of the sampling.

*Candida sp* was isolated in 67 samples (39.5%). A complete identification of this *Candida* would have provided additional information on the *Candida* species circulating in Senegal.

## 5. Conclusions

This study confirm that onychomycosis is a public health concern in health facilities in Dakar. *Candida albicans* and *Trichophyton soudanense* are the most common agents of onychomycosis. The diagnostic cannot only be based on the clinical aspect of the lesions. The place of mycological exams is very important in confirming the diagnostic, identifying the fungal agent and guiding the treatment. By characterizing the spectrum of pathogens usually involved, the results of this study could contribute to improve the management of onychomycosis.

## Figures and Tables

**Figure 1 jof-05-00035-f001:**
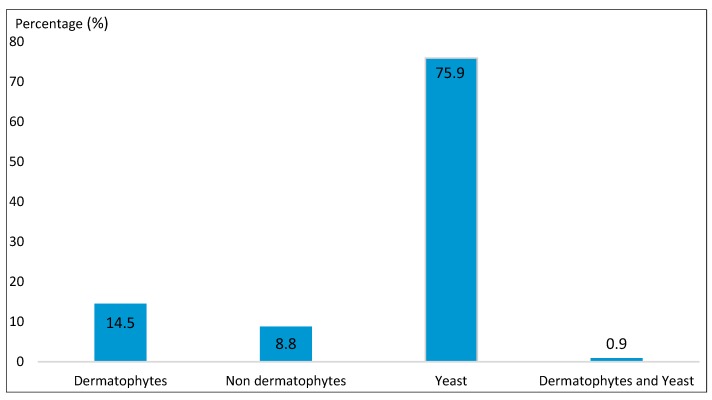
Identified pathogens groups.

**Table 1 jof-05-00035-t001:** Socio-demographic characteristics of the study population (*n* = 469).

Parameters	Number	Percentage (%)	95% CI
**Age group**			
<15 years	49	10.5	7.7–13.8
>15 years	420	89.5	81.2–98.5
**Gender**			
Female	309	65.9	58.74–73.6
Male	160	34.1	29.1–39.8
**Season**			
Dry	351	74.8	67.2–83.1
Rainy	118	25.2	20.8–30.1

**Table 2 jof-05-00035-t002:** Clinical aspect of the lesions (*n* = 469).

Clinical Characteristic	Number	Percentage (%)	95% CI
DLSO	178	37.9	32.6–44
WSO	15	3.2	1.8–5.3
Onyxis	218	46.5	40.5–53.1
Perionyxis	11	2.4	1.2–4.2
Onyxis/Perionyxis	47	10	7.4–13.3

DLSO: Disto-Lateral Subungual Onychomycosis, WSO: White Superficial Onychomycosis.

**Table 3 jof-05-00035-t003:** Isolated fungal species (*n* = 227).

	Number	Percentage (%)	95% CI
**Dermatophytes**			
*Microsporum langeronii*	1	0.4	0.01–2.4
*Trichophyton mentagrophytes*	1	0.4	0.01–2.4
*Trichophyton rubrum*	3	1.3	0.2–3.8
*Trichophyton soudanense*	23	10.1	6.4–15.2
*Trichophyton verrucosum*	1	0.4	0.01–2.4
*Trichophyton violaceum*	3	1.3	0.2–3.8
*Trichophyton asahii*	1	0.4	0.01–2.4
**Non-dermatophytes**	20	8.8	
*Acremonium* sp.	1	0.4	0.01–2.4
*Aspergillus candidus*	1	0.4	0.01–2.4
*Aspergillus niger*	1	0.4	0.01–2.4
*Fusarium solani*	1	0.4	0.01–2.4
*Fusarium* sp.	12	5.3	2.7–9.2
*Onychocola* sp.	2	0.9	0.1–3.2
*Scedosporium* sp.	1	0.4	0.01–2.4
*Scytalidium* sp.	1	0.4	0.01–2.4
**Yeast**			
*Candida albicans*	97	42.7	34.6–52.1
*Candida glabrata*	1	0.4	0.01–2.4
*Candida parapsilosis*	1	0.4	0.01–2.4
*Candida tropicalis*	6	2.6	0.9–5.7
*Candida* sp.	67	29.5	22.8–37,5
**Dermatophytes and Yeast**			
*Trichophyton soudanense + Candida albicans*	2	0.9	0.1–3.2

**Table 4 jof-05-00035-t004:** Prevalence of onychomycosis according to the socio-demographic characteristics of the study population (*n* = 227).

Parameters	Number	Percentage (%)	*p* Value
**Age group**			
<15 years	23	46.9	0.83
>15 years	204	48.6	
**Gender **			
Female	158	51.1	0.1
Male	69	43.1	
**Season**			
Dry	172	49	0.65
Rainy	55	46.6	

**Table 5 jof-05-00035-t005:** Frequency of onychomycosis based on the clinical characteristics of the lesions.

Results	DLSO	WSO	Onyxis	Onyxis/Perionyxis	Perionyxis	Total
**Negative**	102 57.3%	4 26.7%	114 52.3%	20 42.5%	2 18.2%	242 51.6%
**Positive**	76 42.7%	11 73.3%	104 47.7%	27 57.5%	9 81.8%	227 48.4%
**Total**	178 100%	15 100%	218 100%	47 100%	11 100%	469 100%

Chi2 = 12.5; *p* value = 0.014, DLSO: Disto-Lateral Subungual Onychomycosis, WSO: White Superficial Onychomycosis.

**Table 6 jof-05-00035-t006:** Variation of Onychomycosis caused by *Candida albicans* and *Trichophyton soudanense* according to the socio-demographic characteristics of patients.

	Total Positive	Number of *C. albicans*	Percentage (%)	95% CI	Number of *T. soudanense*	Percentage (%)	95% CI
**Age group**							
<15 years	23	8	34.8	15–68.5	4	17.4	4.7–44.5
>15 years	204	89	43.6	35–53.7	19	9.7	5.6–14.5
**Gender**							
Female	158	71	45	35.1–56.7	13	8.2	4.4–14.1
Male	69	26	37.7	24.6–55.5	10	14.5	6.9–26.6
**Season**							
Dry	172	77	44.8	35.3–56	17	10	5.7–15.8
Rainy	55	20	36.4	22.2–56.2	6	11	0.4–23.7

**Table 7 jof-05-00035-t007:** Variation of Onychomycosis with *Candida albicans* and *Trichophyton soudanense* according to the clinical aspects of the lesions.

Clinical Characteristic	Total Positive	Number of *C. albicans*	Percentage (%)	95% CI	Number of *T. soudanense*	Percentage (%)	95% CI
DLSO	76	33	43.4	30–61	6	7.9	2.9–17.2
WSO	11	1	9.1	0.2–50.6	1	9.1	0.2–50
Onyxis	104	40	38.5	27.5–52.3	15	14.4	8.9–23.7
Perionyxis	9	5	55.6	18.1–99.9	1	11.1	0.2–61.9
Onyxis/Perionyxis	27	18	66.7	39.6–99.9	--	--	--

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
