# Peer review of "Epidemiological and Mycological Aspects of Onychomycosis in Dakar (Senegal)"

_jof, 2019, doi:10.3390/jof5020035_

Round 1
Reviewer 1 Report
line 38--it is aesthetic for the patient, and then you go on to say what complications it can cause. I would rephrase to say it is aesthetic from the patient's point of view.
the discussion is right on target
Author Response
Reviewer Comment :
Reviewer #1:
1. Reviewer comment: Moderate English changes required
Response:The manuscript has been rewritten and all necessary correction was done.
2. Reviewer comment: Line 38--it is aesthetic for the patient, and then you go on to say what complications it can cause. I would rephrase to say it is aesthetic from the patient's point of view
Response:We agree with the reviewer, all necessary correction was done
Line 38-- The consequences of onychomycosis are mainly aesthetic from the patient’s point of view
Reviewer 2 Report
I presume the authors mean potassium hydroxide as the medium for direct microscopy ?
Onyxis means ingrowing nails – is this what the authors were seeing in these patients – it seems quite a high number.
I was also a bit surprised by the low numbers of cases of paronychia or perionyxis
It would be important that the authors explain how they define Candida species as nail pathogens here. The problem with this organism is that it can be a nail commensal as well as a pathogen but there are some methods of trying to establish its role such as repeat isolations , the presence of hyphae in direct microscopy or relate it to the clinical appearances. I note that some came from DLSO and some from onycholysis – how did you define each ? This is all important as otherwise the paper is simply describing the fungal flora of the nails rather than the causes of onychomycosis.
The absence of Neoscytalidium is worth pointing out given that it has been described in West Africa as a common cause of skin infection on the feet.
Author Response
Reviewer #2:
1 . Reviewer comment: Moderate English changes required
Response:The manuscript has been rewritten and all necessary correction was done.
2. Reviewer comment: I presume the authors mean potassium hydroxide as the medium for direct microscopy ?
Response:We agree with the reviewer, the correction was done
Line 107-- A proportion of the collected nails samples was digested by 30% potassium hydroxide (KOH).
3. Reviewer comment:I was also a bit surprised by the low numbers of cases of paronychia or perionyxis
Response:Onyxis or onychia mean ingrowing nails / Paronychia or perionyxis
The number of cases of onychia and paronychia in this study is 218 (46,5%) and 11 (2,4%). This has been described by other studies.
Seck, MC, Ndiaye, D, Diongue K, Ndiaye M, Badiane AS, Sow D, et al. Profil mycologique des onychomycoses à Dakar (Sénégal). J Myco Med 2014 ; 24:124-128.
Konate A, Yavo W, Kassi KF, Djohan V, Angora KE, Bosson-Vanga H, et al. Profil mycologique des onychomycoses à Abidjan (Côte d’Ivoire). J Myco Med, 2014 ; 24: 205-210.
4. Reviewer comment: It would be important that the authors explain how they define Candida species as nail pathogens here. The problem with this organism is that it can be a nail commensal as well as a pathogen but there are some methods of trying to establish its role such as repeat isolations, the presence of hyphae in direct microscopy or relate it to the clinical appearances.
Response:In addition to macroscopic and microscopic aspects, confirmation of Candida species was based on observation of pseudomycelium under light microscopy, germ tube test, chlamydo-spore formation and the AUXACOLOR rapid identification system (Biorad, France).
5. Reviewer comment: I note that some came from DLSO and some from onycholysis – how did you define each ?
Response:
DLSO: is defined by invasion of the nail bed and underside of the nail plate.
The correct word to be used is “white superficial onychomycosis” not
onycholysis.
White superficial onychomycosis (WSO): is a less common variety, is a distinctive pattern in which the nail plate is the primary site of invasion. This is the surface infection of the nail primarily when the fungi invade the superficial layers of the nail plate directly
5. The absence of Neoscytalidium is worth pointing out given that it has been described in West Africa as a common cause of skin infection on the feet.
In our study we did not find Neoscytalidiumeven if this species is found in West Africa.
Round 2
Reviewer 2 Report
The authors have responded to my queries
1. My original point about the use of the word onyxis was simply to clarify that they meant ingrowth of the nail. If that is correct its is preferable to use onyxis than onychia, a term which is now rarely used
So if this is ingrowth of the nail plate the first time the word is used the authors should simply write onyxis followed by ( ingrowth of the nail plate) in brackets. This then clarifies what they mean.
2. In this section line 265 " such as Onychocola sp and Scytalidium sp were also isolated in ". I presume they mean were also not isolated . It should also be Neoscytalidium
Author Response
Reviewer #2:
Reviewer comment:
My original point about the use of the word onyxis was simply to clarify that they meant ingrowth of the nail. If that is correct its is preferable to use onyxis than onychia, a term which is now rarely used
So if this is ingrowth of the nail plate the first time the word is used the authors should simply write onyxis followed by (ingrowth of the nail plate) in brackets. This then clarifies what they mean.
Response:The common word used is onyxis.
It’s preferable to use onyxis defined as an ingrowing of nail.
Necessary correction was done in the main manuscript.
reviewer comment:
In this section line 265 " such as Onychocola sp and Scytalidium sp were also isolated in ". I presume they mean were also not isolated. It should also be Neoscytalidium
Response:
In line 315, we said that “Pseudo-dermatophytes such as Onychocola sp (0.4%)and Scytalidium sp (0.4%)were also isolated in this study”.
It was what our results described (Table 3. Isolated fungal species (n=227))
Several studies conducted in Africa (Marocco, Gabon, Cameroon) have demonstrated the presence of Scytalidium dimidiatum and as Onychocola sp.
Hajoui, F. Z., et al. The mould onychomycosis in Morocco: about 150 isolated cases in 20 years. J Myco Med, 2012; 22:3, 221-224.
Afène SN, Ngoungou EB, Mamfoumbi MM, Akotet MB, Mba IA, Kombila M. Les onychomycoses au Gabon: aspects cliniques et mycologiques. J Myco Med 2011; 21 :248-255.
Nkondjo Minkoumou, Salvador, Valentina Fabrizi, and Manuela Papini. "Onychomycosis in Cameroon: a clinical and epidemiological study among dermatological patients." International journal of dermatology 51.12 (2012): 1474-1477.
RegardingScytalidium sp, it was described that Coelomycetes of the genus Scytalidiuminclude several species such as Scytalidium lignicola(currently called Scytalidium dimidiatum, by Pesante 1957) and Scytalidium hyalinum. Reference 1 & 2
Recent classification : Scytalidiumdimidiatumis also called Neoscytalidium dimidiatum(Reference 3)
References :
XAVIER, Ana Paula Martins, OLIVEIRA, Jeferson Carvalhaes de, RIBEIRO, Vera Lúcia da Silva, et al. Epidemiological aspects of patients with ungual and cutaneous lesions caused by Scytalidium spp. Anais brasileiros de dermatologia, 2010, vol. 85, no 6, p. 805-810.
Lacaz CD, Pereira AD, Heins-Vaccari EM, Cucé LC, Benatti C, Nunes RS, FREITAS-LEITE RS, HERNÁNDEZ-ARRIAGADA GL. Onychomycosis caused by Scytalidium dimidiatum. Report of two cases. Review of the taxonomy of the synanamorph and anamorph forms of this coelomycete. Revista do Instituto de Medicina Tropical de São Paulo. 1999 Sep;41(5):318-23.
Tan DH, Sigler L, Gibas CF, Fong IW. Disseminated fungal infection in a renal transplant recipient involving Macrophomina phaseolina and Scytalidium dimidiatum: case report and review of taxonomic changes among medically important members of the Botryosphaeriaceae. Medical Mycology. 2008 May 1;46(3):285-92.
XAVIER, Ana Paula Martins, OLIVEIRA, Jeferson Carvalhaes de, RIBEIRO, Vera Lúcia da Silva, et al. Epidemiological aspects of patients with ungual and cutaneous lesions caused by Scytalidium spp. Anais brasileiros de dermatologia, 2010, vol. 85, no 6, p. 805-810.